# Physical activity, fear avoidance beliefs and level of disability in a multi-ethnic female population with chronic low back pain in Suriname: A population-based study

Nancy Ho-A-Tham[1,2]*, Niels Struyf[1,2], Beverly Ting-A-Kee[3], Johanna de Almeida Mello[4], Yves Vanlandewijck[5,6], Wim Dankaerts[2]

1 Department of Physiotherapy, Faculty of Medical Sciences, Anton de Kom University of Suriname, Paramaribo, Suriname, 2 Department of Rehabilitation Sciences, Research Group for Musculoskeletal Rehabilitation, Faculty of Movement and Rehabilitation Sciences, KU Leuven, Leuven, Belgium, 3 Department of Pathology, Faculty of Medical Sciences, Anton de Kom University of Suriname, Paramaribo, Suriname, 4 LUCAS, Center for Care Research and Consultancy, KU Leuven, Leuven, Belgium, 5 Department of Rehabilitation Sciences, Research Group of Adapted Physical Activity and Psychomotor Rehabilitation, Faculty of Movement and Rehabilitation Sciences, KU Leuven, Leuven, Belgium, 6 Department of Physiology, Nutrition and Biomechanics, Swedish School of Sport and Health Sciences, Stockholm, Sweden

* nancy.ho-a-tham@uvs.edu, nancyhoatham@gmail.com

## Abstract

### Background

Chronic low back pain (CLBP) is an important cause for reduced daily physical activity (PA) and loss of quality of life, especially in women. In Suriname, a middle-income country in South America, the relationship between PA and CLBP is still unknown.

### Aims

To assess the level of PA in women with CLBP of different ethnicity, and to identify whether fear avoidance beliefs (FAB), disability, co-occurring musculoskeletal pain sites and various sociodemographic and lifestyle factors were associated with self-reported PA.

### Methods

A cross-sectional community-based house-to-house survey was conducted between April 2016 and July 2017. The survey followed the Community Oriented Program for Control of Rheumatic Diseases methodology. Selection criteria were being female of Asian-Surinamese, African-Surinamese or of Mixed ethnicity and aged 18 or older, living in an urban area, and reporting CLBP. Data was collected on PA, FAB, disability, co-occurring musculoskeletal pain sites, CLBP intensity and sociodemographic and lifestyle factors.

### Results

Urban adult women with current CLBP (N = 210) were selected. Nearly 57% of the population met the WHO recommendation on PA, with work-related PA as the largest contributor

**Data Availability Statement:** Data used in this paper cannot be shared publicly due to restrictions

imposed by the Ethics Committee "Commissie Mensgebonden Wetenschappelijk Onderzoek" of the Ministry of Health in Suriname. Proposals to access data from this study can be submitted to the Faculty of Medical Sciences, Anton de Kom University of Suriname (contact bureau-fmew@uvs.edu), to the corresponding author or to Beverly Ting-A-Kee (contact Beverly.ting-a-kee@uvs.edu) and the data will be made available for researchers who meet the criteria for access to confidential data.

**Funding:** Data collection for the COPCORD survey was funded by a grant from the International League of Associations for Rheumatology project 2015 and additional support was received from the Anton de Kom University of Suriname-VLIR-UOS project. The content is solely the responsibility of the authors and does not necessarily represent official views of the funding organizations. The funders had no role in study design, data collection and analysis, decision to publish, or preparation of the manuscript.

**Competing interests:** The authors have declared that no competing interests exist.

to total self-reported PA. Most women showed low FAB scores (FABQ-Work ≤34 (96.2%) and FABQ-PA ≤14 (57.6%)) and low disability levels (Oswestry Disability Index ≤20 (62.4%)). An inverse association between total PA and FABQ-Work (OR = 0.132, CI: 0.023; 0.750) was found. In contrast, total PA had a significant, positive association with disability (OR = 2.154, CI: 1.044; 4.447) and workload (OR = 2.224, CI: 1.561; 3.167). All other variables showed no association with total PA.

## Conclusion

This was the first study in Suriname reporting that 43.3% of urban adult women with CLBP were physically inactive. Total self-reported PA is influenced by FABQ-Work, average to heavy workload and moderate to severe disability. In this study, PA-Work was the major contributor to total PA. Therefore, future longitudinal studies should evaluate different types and aspects of PA in relation to CLBP management.

## Introduction

Epidemiological studies on the prevalence of chronic low back pain (CLBP) have consistently shown that low back pain (LBP) is a relevant health problem, especially in adult females [1]. CLBP is associated with reduced daily physical activity (PA) and loss of quality of life [2, 3]. PA is described by the World Health Organization (WHO) as "*any bodily movement produced by skeletal muscles that requires energy expenditure*" [4]. Insufficient PA and sedentary lifestyle have been reported as risk factors for LBP, especially in older adults [3, 5, 6].

The health benefits of PA or PA-based interventions for chronic musculoskeletal pain are well documented. PA-based interventions appear to be effective in reducing pain and disability in patients with chronic musculoskeletal pain conditions [7, 8]. An inverse relationship between PA and LBP prevalence has been reported [9]. Even when CLBP is already present, moderate to vigorous leisure time PA leads to less pain and disability [10, 11]. For LBP, regular PA has a significant protective effect [10, 12]. The protective effect of leisure time PA on CLBP is seen in both men and women, as well as in adults or older people. However, it has also been reported that individuals with LBP may limit their leisure time PA due to fear of pain [13]; this may progress to a vicious circle of neurological pain adaptations, and consequently lead to disability [14]. Several studies have shown various PA levels within LBP populations. In persons with acute or subacute LBP the level of PA seems to be independent of their pain-related disability [15]. While persons with CLBP and high levels of disability are likely to have low levels of PA [15, 16].

Nowadays, most clinical guidelines on CLBP management, recommend gradual physical reactivation and avoidance of bed rest. However, these recommendations might be difficult to follow for individuals with CLBP, if they possess pessimistic beliefs and attitudes regarding movement or any form of PA [13]. As fear avoidance beliefs (FAB) may lead to fear avoidance behavior, individuals with LBP can become fearful and avoidant of PA due to their pain experience or fear of reinjury [13]. FAB are one of the most important psychological factors for predicting disability in individuals with CLBP [17, 18]. In patients with CLBP, high FAB about work have been reported as a predictor for long-term sick leave, disability, and pain [19].

Several studies suggest that the presence of comorbidities can affect health related quality of life and symptom severity significantly [20, 21]. It is therefore imperative to consider

comorbidities in LBP management as they may act as barriers for PA [22]. A possible comorbidity that may act as a barrier could be co-occurring musculoskeletal pain (co-MSP). Co-MSP is common in CLBP and could influence the prognosis [23–25]. Previous studies indicated that number of chronic pain sites are inversely associated with recovery from CLBP [25] and that presence of co-MSP in other body regions may worsen the prognosis of CLBP [26].

Much attention has been given in literature to review the relationship between PA and FAB [13, 27–29]. But although it has been reported that there are racial, ethnic, and cultural discrepancies in pain beliefs, cognitive factors, and behaviors in different populations for chronic pain [30], evidence regarding PA and FAB in a population with different ethnic origins is scarce. The multi-ethnic society of Suriname, a middle-income country in South-America, provides a unique opportunity to compare ethnic differences in the relationship between PA and FAB. A previous study in Suriname showed that there is a high prevalence of LBP in the urban population, especially in adult females [31]. However, little is known regarding the level of PA and its association with various factors including FAB, level of disability and sociodemographic factors in these women. The aims of the present study were therefore (1) to assess the level of PA in women with CLBP of different ethnicity and (2) to identify whether FAB, level of disability, co-MSP sites and various sociodemographic and lifestyle factors were associated with the level of PA. In the context of CLBP management, it is important to investigate and understand these variables.

## Material & methods

### Study design and participants

A cross-sectional community-based survey with the Community Oriented Program for Control of Rheumatic Diseases (COPCORD) methodology, stage 1: phase 1 and 2, was executed between April 2016 and July 2017 in two urban districts (Paramaribo and Wanica) in Suriname. The methodology and the sampling method of this current study was in line with the COPCORD Suriname study which has been published previously [32]. In this study, the first two phases of stage I from the COPCORD methodology were conducted. In phase 1, information about sociodemographic variables (including occupation and education), presence of musculoskeletal complaints, and medical and trauma history was collected. Participants continued with phase 2 (at the same visit) if they reported any current or past musculoskeletal complaints. In phase 2 additional data regarding CLBP was collected. The original COPCORD questionnaire was translated and adapted for the local population and complemented with two standard questionnaires regarding LBP beliefs. Both phase 1 and 2 questionnaires were administered by interviewers trained by a rheumatologist (RW). The target number of household addresses (N = 1125) in our study was based on an average number of four respondents per household and on population proportions from the Census data of 2012. A random sample of household addresses within different areas of the two districts were selected via a stratified, multi-stage, cluster, sampling design as described by Krishnadath et al. [33]. A door-to-door survey was conducted afterwards. Maximum three attempts were made to interview each eligible household member.

The data used in the study is from a parent (COPCORD Suriname) survey, with a total population of 2221 [32]. For the present study we included all women from the COPCORD urban study population who reported experiencing CLBP (i.e., pain, muscle tension or stiffness between the costal margin and the lower gluteal folds for 12 weeks or longer), who were ≥18 years and of Asian-Surinamese, African-Surinamese or Mixed ethnicity. Written informed consent was obtained from all participants and confidentiality was strictly maintained.

Ethical approval was granted by the Ethics Committee of the Ministry of Health in Suriname (Approval number VG016-14).

## Ethnicity

In Suriname, the five largest ethnic groups are represented by Hindustani (South Asian descendants mainly from India), Javanese (descendants of immigrants from Java, Indonesia), Creoles (African descendants that remained in the city during slavery), Maroons (African descendants that escaped into the hinterland during slavery), and individuals with mixed ethnicity. For this study eligible participants were categorized in Asian-Surinamese (Hindustani and Javanese), African-Surinamese (Creoles and Maroons), and Surinamese of mixed ethnicity. Ethnicity was self-reported via grandparental origins [33]. A person was categorized into a particular ethnic group if at least three of the four grandparents were of the same ethnicity of that specific group. All others were categorized as mixed ethnicity.

## Outcome measures

Information on participants' characteristics such as age, level of education, relationship status, perceived physical workload, LBP intensity and the presence of comorbidities were obtained using the COPCORD questionnaires [32]. To calculate the body mass index (BMI) and to assess waist circumference, weight, height, and waist circumference were measured during the visit.

For PA assessment we used the Global Physical Activity Questionnaire (GPAQ), developed by the WHO. According to Bull et al. [34], reliability coefficients were of moderate to substantial strength (Kappa 0.67 to 0.73; Spearman's rho 0.67 to 0.81). Results on concurrent validity between the International Physical Activity Questionnaire, a previously validated and accepted measure of PA, and GPAQ also showed a moderate to strong positive relationship (range 0.45 to 0.65). The GPAQ demonstrated fair-to-moderate correlations (between the GPAQ and accelerometer) for moderate to vigorous PA when used in face-to-face interviews ($r_s = 0.46$) [35]. The GPAQ gathers information about the frequency (days) and time (minutes/hours) spent on PA in the last seven days in three domains: work-related (PA-Work), transportation and leisure time PA (PA-Leisure time) [36]. The questionnaire was previously translated in Dutch and pre-tested for reliability and face validity by Baldew et al. [37]. For scoring, the amount of metabolic equivalents (METs)-minutes/week was calculated for each domain. Total self-reported PA, calculated as the sum across all three domains, was classified in low, moderate, and high intensity level as described by the GPAQ analysis framework [36]. For adults 18–64 years, the WHO recommends doing at least 150–300 minutes of moderate intensity PA or 75–150 minutes of vigorous intensity PA throughout the week; or an equivalent combination of both throughout the week [38]. In our study we used the WHO recommended level as a cut-off score to categorize individuals into physically active (moderate to high intensity level) versus inactive (low intensity level) adults.

The fear avoidance beliefs questionnaire (FABQ) was specifically developed for application in CLBP patients [39]. The FABQ consists of 16 items divided into the 'beliefs and fear at work' subscale (FABQ-Work; range 0–42 points) and the 'beliefs and to do PA' subscale (FABQ-PA; range 0–24 points) [39]. Participants replied on a 7-point Likert scale, from 0 (i.e., totally disagree) to 6 (i.e., completely agree). Thresholds of >14 (FABQ-PA) [40] and >34 (FABQ-Work) [41] indicate elevated FAB. It has been shown to have excellent test-retest reliability (ICC = 0.90 for FABQ-PA and ICC = 0.96 for FABQ-Work) [42].

For perceived disability, the Oswestry Disabilty Index (ODI) questionnaire was used, which consists of 10 items [43]. The total score ranges from 0 to 50. To obtain the ODI score the total

sum is multiplied by two, resulting in a possible total score from 0% (no to minimal disability) to 100% (maximal disability). Due to small numbers of participants the scores are stratified as follows: 0–20, no to minimal disability; ≥21, moderate to severe disability. The ODI question-naire has a good construct validity [44], acceptable internal consistency (Cronbach α ranges from .71 to .87) and high test-retest reliability (values range from r = 0.83 to 0.99) [45, 46].

For LBP intensity, the assessment was performed using the numerical rating scale for pain (NRS-11) where the respondents were instructed to choose a single number (from 0 (no pain) to 10 (worst imaginable pain)) that best indicates their level of pain in the past seven days [47]. The NRS-11 is the most widely used scale for the assessment of self-reported pain [47, 48] and it has been shown to be reliable and valid measure of pain in LBP [47–49]. Total number of chronic co-MSP sites (0–2 pain sites, ≥3 pain sites) was determined based on the response to body area(s) affected by chronic pain via a mannequin [50].

## Statistical analysis

Descriptive statistical analyses (proportions) were used to describe demographic characteristics and participants' self-reported data for pain, FAB and the level of disability for the different ethnic groups. The median and interquartile range were calculated to report prevalence of PA in METs-minutes. Non-parametric tests of significance (Kruskal Wallis test) were used for comparison of PA level for the different ethnic groups. Chi-squared tests were used to compare categorical data. To identify significant determinants associated with PA and its specific domains (work-related and leisure time), logistic regression models were constructed. The first model included total self-reported PA as dependent variable. The second model included PA Work domain as dependent variable and the third model used PA Leisure time domain. All three models included all other variables as independent variables. Covariates such as age, relationship status and education were also used in the three models. Validated cut-off points were used to dichotomize variables. Assessments with missing values for any variable were not included in the analysis. For all statistical tests, significance level was set at $p < 0.05$. Data entry and analysis were carried out using SPSS 25 and Stata 14.1.

## Results

### Characteristics of women with chronic low back pain

A total of 224 women met the inclusion criteria to participate in the study. Due to missing data to calculate total PA scores, 14 women were excluded. Our total study population consisted of 210 women with CLBP, with a mean age of 48.7±16.9 years. About 55% of the respondents were single and 144 out of 210 women reported doing work activities that were experienced as average or heavy during the day. Most women were overweight or obese with a waist circumference of at least 80 cm (86.2%). A majority of the respondents reported an NRS-11 pain score of 3 or higher (84.3%), and an ODI of ≤20% (62.4%) (Table 1). Most of the women in the present study showed low FAB due to CLBP, reporting scores below the cutoff point for both the FABQ-Work ≤34 (96.2%) and FABQ-PA ≤14 (57.6%) subscales (Table 1).

### Physical activity

Nearly 57% of the population met the WHO recommendation for PA (moderate to high level of PA) (Table 2). No statistical difference was observed between the various ethnic groups for their level of PA (p = 0.854). Although not significant, PA at work contributed largely to the total PA score compared to leisure time activities and transport across all ethnic groups (Table 2).

**Table 1. Descriptive data of participants with chronic low back pain (CLBP).**

| | Total N = 210 | Asian-Surinamese n = 96 | African-Surinamese n = 58 | Mixed n = 56 |
|---|---|---|---|---|
| **Age[a]** | | | | |
| <60 | 150 (71.4%) | 65 (67.7%) | 45 (77.6%) | 40 (71.4%) |
| ≥60 | 60 (28.6%) | 31 (32.3%) | 13 (22.4%) | 16 (28.6%) |
| **Relationship status** | | | | |
| single | 116 (55.2%) | 42 (43.8%) | 39 (67.2%) | 35 (62.5%) |
| in a relationship | 94 (44.8%) | 54 (56.2%) | 19 (32.8%) | 21 (37.5%) |
| **Education[b]** | | | | |
| none or primary | 103 (49.0%) | 65 (67.7%) | 24 (41.4%) | 14 (25.0%) |
| secondary or tertiary | 107 (51.0%) | 31 (32.3%) | 34 (58.6%) | 42 (75.0%) |
| **Work activities (incl. household activities)** | | | | |
| no | 41 (19.5%) | 14 (14.6%) | 17 (29.3%) | 10 (17.9%) |
| yes | 169 (80.5%) | 82 (85.4%) | 41 (70.7%) | 46 (82.1%) |
| **Perceived physical workload** | | | | |
| none to light | 66 (31.4%) | 33 (34.4%) | 20 (34.5%) | 13 (23.2%) |
| average or heavy | 144 (68.6%) | 63 (65.6%) | 38 (65.5%) | 43 (76.8%) |
| **Body Mass Index (BMI)[c]** | | | | |
| normal | 40 (19.0%) | 6 (6.3%) | 16 (27.6%) | 18 (32.1%) |
| overweight | 72 (34.3%) | 37 (38.5%) | 17 (29.3%) | 18 (32.1%) |
| obese | 98 (46.7%) | 53 (55.2%) | 25 (43.1%) | 20 (35.7%) |
| **Waist circumference** | | | | |
| <80 cm | 29 (13.8%) | 11 (11.5%) | 11 (19.0%) | 7 (12.5%) |
| ≥80 cm | 181 (86.2%) | 85 (88.5%) | 47 (81.0%) | 49 (87.5%) |
| **Numeric rating scale for pain** | | | | |
| 1–2 | 33 (15.7%) | 11 (11.5%) | 9 (15.5%) | 13 (23.2%) |
| ≥3 | 177 (84.3%) | 85 (88.5%) | 49 (84.5%) | 43 (76.8%} |
| **Number of co-occurring pain sites** | | | | |
| 0–2 | 138 (65.7%) | 59 (61.5%) | 40 (69.0%) | 39 (69.6%) |
| ≥3 | 72 (34.3%) | 37 (38.5%) | 18 (31.0%) | 17 (30.4%) |
| **Fear Avoidance Beliefs Questionnaire** | | | | |
| FABQ-Work ≤34 | 202 (96.2%) | 89 (92.7%) | 58 (100.0%) | 55 (98.2%) |
| FABQ-Work >34 | 8 (3.8%) | 7 (7.3%) | 0 (0.0%) | 1 (1.8%) |
| FABQ-PA ≤14 | 121 (57.6%) | 50 (52.1%) | 35 (60.3%) | 36 (64.3%) |
| FABQ-PA >14 | 89 (42.4%) | 46 (47.9%) | 23 (39.7%) | 20 (35.7%) |
| **Oswestry Disability Index[d]** | | | | |
| no to minimal disability (ODI ≤20) | 131 (62.4%) | 48 (50.0%) | 40 (68.9%) | 43 (76.8%) |
| moderate to severe disability (ODI >20) | 78 (37.1%) | 48 (50.0%) | 17 (29.3%) | 13 (23.2%) |

[a]in Suriname retirement age is 60 years;

[b]None: no formal education completed, Primary: kindergarten through sixth grade, Secondary: between primary and tertiary education, Tertiary: higher education leading to academic degree;

[c]African-Surinamese and other ethnicities: overweight BMI ≥25 kg/m$^2$ and obesity as BMI ≥30 kg/m$^2$, Asian-Surinamese: overweight BMI ≥23 kg/m$^2$ and obesity as BMI ≥27.5 kg/m$^2$;

[d]missing n = 1

## Regression analysis

The logistic regression model as shown in Table 3 demonstrated an inverse association between the recommended self-reported PA and FABQ-Work (OR = 0.132, CI: 0.023; 0.750),

**Table 2. Self-reported physical activity (PA) in a typical week.**

|  | Asian-Surinamese n = 96 | African-Surinamese n = 58 | Mixed n = 56 | Total N = 210 | P-value |
|---|---|---|---|---|---|
| **Number (n (%)) of women for each level PA[a]** | | | | | |
| Low | 43 (44.8) | 26 (44.8) | 22 (39.3) | 91 (43.3) | .696[b] |
| Moderate | 12 (12.5) | 9 (15.5) | 14 (25.0) | 35 (16.7) | .132[b] |
| High | 41 (42.7) | 23 (39.7) | 20 (35.7) | 84 (40.0) | .775[b] |
| **Median (IQR) of total PA METs-minutes per week** | | | | | |
|  | 2,240 (0–8,670) | 1,950 (110–5,050) | 1,680 (165–8,040) | 1,920 (55–7,830) | 0.854[c] |
| **Median (IQR) of the domain specific contribution to total PA** | | | | | |
| Work | 50.0 (0.0–99.7) | 54.95 (0.0–92.2) | 50.0 (0–95.9) | 50.0 (0.0–96.5) | .871[c] |
| Transport | 0.0 (0.0–18.3) | 3.19 (0.0–35.2) | 0.0 (0.0–29.8) | 0.0 (0.0–23.1) | .271[c] |
| Leisure time | 0.0 (0.0–0.0) | 0.0 (0.0–5.84) | 0.0 (0.0–4.44) | 0.0 (0.0–0.0) | .159[c] |

[a]High: Total PA METs min/week ≥1500 and vigorous PA ≥3 days, OR Total PA METs min/week ≥3000 and vigorous or moderate PA ≥7 days; Moderate: Total PA METs min/week ≥600 AND vigorous PA ≥3 days and ≥ 60 min, OR vigorous to moderate PA ≥5 days and ≥150 minutes days, OR vigorous to moderate PA ≥5 days; Low: value is below high to moderate levels of PA;

[b]Chi square;

[c]Independent Kruskall-Wallis test;

METs = Metabolic Equivalents; IQR = Interquartile Range

which means that a higher perception of FAB is associated with lower total PA time, when compared with people with less FAB. In contrast, a significant and strong positive association between recommended WHO PA level and both the ODI score (OR = 2.154, CI: 1.044; 4.447) and perceived physical workload (OR = 2.223, CI: 1.561; 3.167) was found. No significant association between PA and co-MSP sites was found (Table 3).

**Table 3. Regression analysis for Physical Activity (PA) and its domains.**

|  | Total self-reported PA[#] | | | | PA-Work domain[#] | | | | PA-Leisure time domain[#] | | | |
|---|---|---|---|---|---|---|---|---|---|---|---|---|
| **Independent variables** | **Odds Ratio** | **P>z** | **[95% Conf. Interval]** | | **Odds Ratio** | **P>z** | **[95% Conf. Interval]** | | **Odds Ratio** | **P>z** | **[95% Conf. Interval]** | |
| Age ≥ 60 | .523 | 0.094 | .245 | 1.116 | 1.024 | 0.950 | .480 | 2.186 | **.374*** | 0.034 | .151 | .929 |
| In a relationship (yes = 1) | .941 | 0.851 | .496 | 1.783 | 1.219 | 0.536 | .650 | 2.289 | .942 | 0.868 | .465 | 1.906 |
| Secondary or tertiary education | .969 | 0.925 | .513 | 1.834 | 1.338 | 0.364 | .714 | 2.509 | **2.099*** | 0.041 | 1.030 | 4.279 |
| Smoke (yes = 1) | 1.664 | 0.183 | .787 | 3.519 | .833 | 0.615 | .409 | 1.698 | 1.623 | 0.220 | .749 | 3.517 |
| Presence of comorbidities | .691 | 0.320 | .333 | 1.432 | .917 | 0.811 | .449 | 1.869 | .596 | 0.173 | .284 | 1.254 |
| Work activities (yes = 1) | .508 | 0.143 | .205 | 1.257 | .701 | 0.424 | .293 | 1.676 | - | - | - | - |
| Workload (average to heavy) | **2.224 *** | 0.000 | 1.561 | 3.167 | **1.891*** | 0.000 | 1.352 | 2.644 | - | - | - | - |
| BMI (overweight or obesity) | .851 | 0.724 | .348 | 2.080 | 1.068 | 0.884 | .442 | 2.583 | 1.305 | 0.591 | .494 | 3.449 |
| Waist circumference (≥ 80 cm) | .590 | 0.327 | .206 | 1.695 | .889 | 0.821 | .322 | 2.454 | .963 | 0.944 | .331 | 2.796 |
| Numeric Rating Scale for pain ≥ 3 | .563 | 0.196 | .235 | 1.346 | **.402*** | 0.046 | .164 | .986 | **.289*** | 0.004 | .124 | .677 |
| Number of co-occurring pain sites (2 or more) | 1.198 | 0.600 | .609 | 2.355 | .939 | 0.852 | .485 | 1.817 | 1.055 | 0.887 | .504 | 2.207 |
| FABQ-Work >34 | **.132 *** | 0.022 | .023 | .750 | **.170*** | 0.037 | .032 | .902 | - | - | - | - |
| FABQ-PA >14 | .789 | 0.491 | .402 | 1.548 | - | - | - | - | 1.146 | 0.727 | .535 | 2.455 |
| ODI >20 (moderate to severe disability present) | **2.154*** | 0.038 | 1.044 | 4.447 | **2.404*** | 0.014 | 1.198 | 4.823 | .588 | 0.211 | .256 | 1.351 |

ODI = Oswestry Disability Index score; BMI = Body Mass Index; FABQ-Work = Fear Avoidance Beliefs Questionnaire subscale work, FABQ-PA = Fear Avoidance Beliefs Questionnaire subscale physical activity; Significance levels:

*** p≤0.000,

**p≤0.01,

*p≤0.05

[#]Recommended WHO moderate to vigorous PA level

In the domain of PA-Work, FAB at work presented a significant and inverse association (OR = 0.170, CI: 0.032; 0.902), which is in concordance with the results for the total level of PA. The level of pain (OR = 0.402, CI: 0.164; 0.986) had an inverse association with PA-Work, while a significant and strong direct association was found with the level of disability (OR = 2.404, CI: 1.198; 4.823) and perceived physical workload (OR = 1.891, CI: 1.352; 2644).

Regarding PA-Leisure time domain, the level of education had a significant and high association (OR = 2.099, CI: 1.030; 4.279), while age ≥60 years and NRS-11 pain scale presented with significant but low OR values (respectively OR = 0.374, CI: 0.151; 0.929 and OR = 0.289, CI: 0.124; 0.677), showing that age and pain have an inverse association with leisure time PA. Higher pain scores were therefore in both domains of the GPAQ a significant determinant for less PA.

## Discussion

This was the first study in Suriname to evaluate the level of PA in women with CLBP addressing possible associations between PA and FAB, level of disability, co-MSP, various sociodemographic and lifestyle factors. The proportion of physically inactive adult women with CLBP in an urban community according to the WHO criteria was 43.3%. Regression analyses showed significant associations between moderate to high levels of total self-reported PA and heavy workload, FABQ-Work and moderate to severe disability. Other variables such as co-MSP sites, level of pain, ethnicity, comorbidities, sociodemographic (age, education, marital status) and lifestyle factors (BMI and waist circumference, smoking, workload, and work activities) showed no association with total PA in this study population.

This population-based cross-sectional study endorses that the prevalence of total PA in the study population of adult women with CLBP who met the WHO recommendations (56.7%) is consistent with previously reported PA numbers for the overall population of Suriname (55.5%) [37]. According to the South American physical activity and sedentary behavior network which also used the GPAQ, Chile and Brazil, reported 51.1% and 52.4% respectively, for PA prevalence in women [51]. Similar prevalence rate for PA was reported for urban women in Ethiopia (50.6%) [52]. However, in our study, PA was evaluated within a specific group of women experiencing CLBP while in other studies data was assessed from the general community via census data [28, 51–53].

The contribution of PA-Leisure time to the total PA score was very low in the present study. In a study in Bangladesh for example, PA-Leisure time by women constituted less than 3% of the total PA [51]. Although it was found that PA-Leisure time may provide modest protection against frequent or CLBP [10, 54], studies have shown that individuals with LBP may limit their leisure-time PA due to fear of pain [11, 55]. In low- and middle-income countries, facilitators for PA may be lacking and social, structural, and individual barriers could be reasons why individuals do not participate in leisure-time PA [56, 57]. Since more than half of the women in our study were single, with no or low education and probably the sole breadwinner in their household, we cautiously hypothesize that poor social and economic status may have led to the necessity to perform more (paid or unpaid) working hours, resulting in low leisure-time PA. Thus, PA during work activities contributed largely to the total PA score in our study. These findings were also found in other low- and middle-income countries [52, 53]. However, PA-Work can increase the risk of recurrent LBP [54]. In the present study, average to heavy workload was associated with total PA and PA-work. High workload has been reported as risk for future LBP [54, 58, 59]. Also, specific types of work activities such as frequent lifting and exposure to high occupational workloads have been proven to harm the

lower back [54, 58]. Consequently, it is necessary that aspects of work such as workload and the type of work activities should be considered when evaluating PA in the context of CLBP management.

In our study, a positive relationship between educational level and PA-Leisure time was reported. Other studies also showed that higher educated individuals were more physically active during leisure-time [6, 60–62]. Persons with a higher educational level might have better resources to incorporate PA in daily lives [62]. In lower educated persons, a possible explanation might be low perceived control of life, an important predictor for decreasing PA-Leisure time [61]. Age was found to be significantly and inversely associated with PA-Leisure time. This association was also found in persons without CLBP [11]. Pain was also a significant determinant for less PA in both PA-Work as in PA-Leisure time. These outcomes are in concordance with the literature [11, 55, 63].

Our study was not in agreement with previous studies investigating the association with PA and disability in CLBP [15, 16]. De Sousa et al. [15] reported an inverse relationship between PA and disability in CLBP patients in Brazil. A systematic review by Lin et al. [16] also reported these same results. In the present study, moderate to severe disability was positively associated with total self-reported PA and PA-work. This means that participants in our study were able to maintain a certain level of PA during work despite reporting moderate to severe levels of disability. Probably, different patterns of PA such as type and quality of PA [64], social desirability leading to overreporting of PA [65] and heterogeneity in the definitions of PA and CLBP may have led to these conflicting results.

Although evidence has shown that an inverse relation is present between daily PA and pain and disability in individuals with CLBP [11, 15, 16], it is debatable whether PA in daily life alone can be effective as an intervention strategy to improve outcome when managing CLPB [9, 66]. For the management of CLBP, specific exercise modalities, as part of PA-Leisure time, and focusing on muscular strength, flexibility and aerobic fitness are recommended in clinical guidelines for CLBP [67–69]. However, these guidelines do not mention which types and intensities of PA will be beneficial in the management of CLBP. Therefore, regular PA combined with patient-specific and structured exercises which are properly progressed should be the cornerstones of the rehabilitation process for LBP [70]. Future studies should focus on interventions combining daily PA with patient-specific exercises in certain subgroups.

As confusing evidence still exist regarding the relationship between PA and CLBP for CLBP management, sedentary behavior might be equally important to pay attention to. Sedentary behavior is defined as 'any waking behaviors characterized by an energy expenditure ≤1.5 METs, while in a sitting, reclining, or lying posture' [71]. According to Thivel et al. [72] a person can be classified as both (in)active and sedentary. In a recent systematic review by Alzahrani et al. [73] sedentary behavior seems to play an important role in the association between CLBP and poor disability. No associations were found between sedentary behavior and LBP risk or pain intensity. This was conflicting evidence with previous findings reporting an association between PA and LBP risk [6, 10, 11]. A combination of sedentary behavior with low levels of PA may increase LBP risk [3]. As PA during work activities contributed largely to the total PA score in our study, it is necessary to include determinants of sedentary behavior during work to develop treatment strategies, especially in occupational health. Sedentary behavior was not within the scope of this study. Future research is needed to elaborate on this topic.

The present study also evaluated the association between PA and FABQ scores among the CLBP women. An inverse relation was found between total PA and FABQ-Work and between the domain PA-Work and FABQ-Work. Literature confirms these findings between PA and FAB [27, 70, 74]. In a recent study by Naugle et al. [75] this inverse association between PA and fear of movement was also seen in chronic pain-free older adults. Various studies,

however, reported no association between PA and FAB [28, 29, 76]. Reasons for these discrepancies can lie in the use of different assessment instruments for PA and FAB. Although FABQ showed good psychometric properties, it does not define the specific activities that are feared or avoided. Leonhardt et al. [29] also stated that possibly the quality and not the quantity of certain activities are associated with FAB. Most women in our study population had overall low FABQ-Work scores and high level of PA-Work. According to Elfving et al. [27] engagement in PA may also modify fear-avoidant beliefs. The need to continue working due to low socio-economic status and lack of workers' compensation in Suriname for CLBP are also possible explanations [77]. Although not within the scope of this study, coping strategies and pain beliefs systems may have led to adaptive responses that promote function. While ethnicity was not a significant factor in our study, adaptive responses on how individuals view CLBP and cope with pain could also be influenced by their culture [78].

It is important that healthcare professionals should be aware of fear avoidance beliefs and behaviors, as it can potentially reduce PA and increase the risk of disability. These maladaptive beliefs together with pain intensity and pain catastrophizing have been reported as important determinants of disability [19, 79]. Recognition of these beliefs are also paramount in the acute or sub-acute phase of LBP, as these unhelpful beliefs can increase the likelihood of delayed recovery and chronicity [80]. Additionally, a reduction in FAB at the early stages of LBP may lead to an increase of activities in daily and social life [79].

## Strengths and limitations of the study

Although no significant differences were found between ethnic groups, the consideration of different ethnic groups is a strength of the current study as it gave us the opportunity to look at the PA levels within these groups. However, due to the small number of the participants, we had to refer to the Javanese and Hindustani as one group throughout the paper. It is important to note that within this group of Asian descendants there are significant language and cultural variations. In our study we also included women who were performing PA related to unpaid or at home working activities. Data on PA was self-reported. Therefore, recall-bias is possible [81]. It is known that respondents are likely to over- or underestimate their engagement in PA [82]. PA should therefore also be monitored objectively via accelerometry in follow-up studies [83]. We did not assess the social, cultural, and environmental factors that may influence PA in these women. Follow-up studies must consider the effects of these factors on PA and should also take into account the different types and aspects of PA (frequency, duration, intensity, and type of PA) for LBP populations. Finally, the cross-sectional nature of our study did not allow us to assess the temporal relation between the assessed factors and PA.

## Conclusion

This was the first study to present findings on the association of PA and CLBP, the most overlooked musculoskeletal public health problem in Suriname. Based on the WHO recommendations for PA, more than half of the study population of adult women with CLBP in an urban community were physically active. In this multi-ethnic cohort of women with CLBP, no association between PA and ethnicity was observed. ODI scores showed a strong association with PA probably due to socio-economic settings and needs, while FABQ scores were inversely related with overall PA and PA-Work. PA related to work activities contributed largely to the total weekly PA. Future longitudinal studies on CLBP in Suriname are needed to evaluate possible influences of socio-economic and work factors (type of work, long hours) in CLBP populations to improve management strategies related to PA.

## Acknowledgments

The authors are grateful to the participants and research team of the COPCORD Suriname study for their contribution.

## Author Contributions

**Conceptualization:** Nancy Ho-A-Tham, Yves Vanlandewijck, Wim Dankaerts.

**Data curation:** Nancy Ho-A-Tham, Niels Struyf, Beverly Ting-A-Kee.

**Formal analysis:** Nancy Ho-A-Tham, Niels Struyf, Beverly Ting-A-Kee, Johanna de Almeida Mello.

**Funding acquisition:** Nancy Ho-A-Tham, Yves Vanlandewijck, Wim Dankaerts.

**Investigation:** Nancy Ho-A-Tham, Niels Struyf, Beverly Ting-A-Kee.

**Methodology:** Nancy Ho-A-Tham, Yves Vanlandewijck, Wim Dankaerts.

**Project administration:** Nancy Ho-A-Tham, Beverly Ting-A-Kee.

**Resources:** Yves Vanlandewijck, Wim Dankaerts.

**Supervision:** Nancy Ho-A-Tham, Yves Vanlandewijck, Wim Dankaerts.

**Validation:** Nancy Ho-A-Tham, Beverly Ting-A-Kee.

**Visualization:** Nancy Ho-A-Tham, Niels Struyf, Beverly Ting-A-Kee, Johanna de Almeida Mello.

**Writing – original draft:** Nancy Ho-A-Tham, Johanna de Almeida Mello, Yves Vanlandewijck, Wim Dankaerts.

**Writing – review & editing:** Nancy Ho-A-Tham, Niels Struyf, Beverly Ting-A-Kee, Johanna de Almeida Mello, Yves Vanlandewijck, Wim Dankaerts.

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
