## [Decision Letter · Decision Letter 0]

9 Jun 2022

PONE-D-21-38825Physical activity, fear avoidance beliefs and level of disability in a multi-ethnic female population with chronic low back pain in Suriname: a population-based studyPLOS ONE

Dear Dr. Ho-A-Tham,

Thank you for submitting your manuscript to PLOS ONE. After careful consideration, we feel that it has merit but does not fully meet PLOS ONE’s publication criteria as it currently stands. Therefore, we invite you to submit a revised version of the manuscript that addresses the points raised during the review process.

We look forward to receiving your revised manuscript.

Kind regards,

Sarah Michiels

Academic Editor

PLOS ONE

Journal Requirements:

Reviewers' comments:

Reviewer's Responses to Questions

**Comments to the Author**

1. Is the manuscript technically sound, and do the data support the conclusions?

Reviewer #1: Partly

Reviewer #2: Yes

2. Has the statistical analysis been performed appropriately and rigorously? 

Reviewer #1: I Don't Know

Reviewer #2: Yes

3. Have the authors made all data underlying the findings in their manuscript fully available?

Reviewer #1: Yes

Reviewer #2: Yes

4. Is the manuscript presented in an intelligible fashion and written in standard English?

Reviewer #1: Yes

Reviewer #2: No

5. Review Comments to the Author

Reviewer #1: Thank you for considering me to review this manuscript entitled “Physical activity, fear avoidance beliefs and level of disability in a multi-ethnic female population with chronic low back pain in Suriname: A population based study”. This study lacks novelty. The rationale is poorly justified in in the introduction. Methodological details are missing. Language could be improved by editing from a native English language editor.

Abstract

Objective seems to be merged into the introduction. Kindly correct it.

Methods: There is no discussion of inclusion/exclusion criteria here which is imperative to include.

Results: Confidence intervals should also be reported along with odds ratio. Interpretation of OR is incomplete without CI.

Authors are discussing associations here which was not mentioned in the objectives of the study. Results should be discussed in line with the objectives of the study.

Introduction

This section could be improved by adding more literature pertaining to the concerned topic. Further, link between PA and other variables should be thoroughly discussed.

Methods

Include information on psychometric properties of the outcomes variables in the methods section.

Categorization of ethnic groups should be clearly mentioned.

Include a paragraph on study protocol where a brief summary of the study can be described.

Statistical analysis

Some points regarding the data analysis are unclear. Intricate details of regression analysis are required. What were the predictors and independent variables in the analysis? What all covariates were considered in the regression analysis?

Discussion

Line 252-254: Age and educational level were not variables of interest. Results should be discussed in line with the objectives of the study.

Discussion seems insufficient. Findings of all variables are not discussed and relevant studies are very scant. A detailed discussion on findings of each variable is required.

Reviewer #2: than you for your effort, and your manuscript needs some revision

The purpose of this study is to examine the physical activity, fear avoidance beliefs and level of disability in a multi-ethnic female population with chronic low back pain in Suriname

The statistical power and research design are sound. Understanding the rationale for the research question can be improved through a revised introduction.

In abstract introduction: Chronic low back pain (CBLP) change to CLBP

Discussion

This section needs more explanation particularly, comparisons with previous similar studies.

6. PLOS authors have the option to publish the peer review history of their article (what does this mean?). If published, this will include your full peer review and any attached files.

Reviewer #1: No

Reviewer #2: No

---

## [Author Response · Author response to Decision Letter 0]

25 Aug 2022

Dear Academic Editor, PLOS ONE, dear Dr. Sarah Michiels, 

We would like to thank you for giving us the opportunity to submit a revised copy of our manuscript entitled ‘Physical activity, fear avoidance beliefs and level of disability in a multi-ethnic female population with chronic low back pain in Suriname: a population-based study’. We appreciate the time and effort that you and the reviewers have dedicated to providing valuable feedback on our manuscript. We are grateful to the reviewers for their insightful comments for correction or modification we believe have resulted in an improved revised manuscript. We have been able to incorporate most of the changes after reflecting on the suggestions provided by the reviewers. We have highlighted these changes within the manuscript via track changes. Please find our responses to reviewer’s remarks, as well as the location of the changes in the revised manuscript and in the document entitled 'Response to reviewers'. We look forward to hearing from you regarding our submission. We would be glad to respond to any further questions and comments that you or the reviewers may have.

Sincerely yours,

Nancy Ho-A-Tham, corresponding author

Dear Reviewer#1,

Thank you for your comments and useful suggestions. We have revised the abstract introduction, to make the rationale of the study clearer to the reader. We have also corrected the sentence of the objectives of the study. Please find our responses to your remarks, as well as the location of the changes in the manuscript and in the document entitled 'Response to reviewers'.

Dear Reviewer#2,

We would like to thank you for your useful comments and suggestions. We appreciate your support. Please find our changes in the manuscript and the document entitled 'Response to reviewers'.

---

## [Decision Letter · Decision Letter 1]

18 Oct 2022

Physical activity, fear avoidance beliefs and level of disability in a multi-ethnic female population with chronic low back pain in Suriname: a population-based study

PONE-D-21-38825R1

Dear Dr. Ho-A-Tham,

We’re pleased to inform you that your manuscript has been judged scientifically suitable for publication and will be formally accepted for publication once it meets all outstanding technical requirements.

Kind regards,

Sarah Michiels

Academic Editor

PLOS ONE

Additional Editor Comments (optional):

Reviewers' comments:

Reviewer's Responses to Questions

**Comments to the Author**

1. If the authors have adequately addressed your comments raised in a previous round of review and you feel that this manuscript is now acceptable for publication, you may indicate that here to bypass the “Comments to the Author” section, enter your conflict of interest statement in the “Confidential to Editor” section, and submit your "Accept" recommendation.

Reviewer #2: All comments have been addressed

2. Is the manuscript technically sound, and do the data support the conclusions?

Reviewer #2: Partly

3. Has the statistical analysis been performed appropriately and rigorously? 

Reviewer #2: Yes

4. Have the authors made all data underlying the findings in their manuscript fully available?

Reviewer #2: Yes

5. Is the manuscript presented in an intelligible fashion and written in standard English?

Reviewer #2: Yes

6. Review Comments to the Author

Reviewer #2: All comments have been addressed for "Physical activity, fear avoidance beliefs and level of disability in a multi-ethnic female population with chronic low back pain in Suriname: a population-based study"

7. PLOS authors have the option to publish the peer review history of their article (what does this mean?). If published, this will include your full peer review and any attached files.

Reviewer #2: No

---

## [Editor Report · Acceptance letter]

21 Oct 2022

PONE-D-21-38825R1 

Physical activity, fear avoidance beliefs and level of disability in a multi-ethnic female population with chronic low back pain in Suriname: a population-based study 

Dear Dr. Ho-A-Tham:

I'm pleased to inform you that your manuscript has been deemed suitable for publication in PLOS ONE. Congratulations! Your manuscript is now with our production department. 

Kind regards, 

on behalf of

Prof. Sarah Michiels 

Academic Editor

PLOS ONE